# DySpec: Faster Speculative Decoding with Dynamic Token Tree Structure

## Abstract

While speculative decoding has recently appeared as a promising direction for accelerating the inference of large language models (LLMs), the speedup and scalability are strongly bounded by the token acceptance rate. Prevalent methods usually organize predicted tokens as independent chains or fixed token trees, which fails to generalize to diverse query distributions. In this paper, we propose DySpec, a faster speculative decoding algorithm with a novel dynamic token tree structure. We begin by bridging the draft distribution and acceptance rate from intuitive and empirical clues, and successfully show that the two variables are strongly correlated. Based on this, we employ a greedy strategy to dynamically expand the token tree at run time. Theoretically, we show that our method can achieve optimal results under mild assumptions. Empirically, DySpec yields a higher acceptance rate and speedup than fixed trees. DySpec can drastically improve the throughput and reduce the latency of token generation across various data distribution and model sizes, which significantly outperforms strong competitors, including Specinfer and Sequoia. Under low temperature setting, DySpec can improve the throughput up to $9.1\times$ and reduce the latency up to $9.4\times$ on Llama2-70B. Under high temperature setting, DySpec can also improve the throughput up to $6.21\times$, despite the increasing difficulty of speculating more than one token per step for draft model.

## 1 Introduction

Recent years have witnessed the prosperity of large language models (LLMs), shown by their unprecedented capabilities in understanding and generating human languages in various domains and tasks (OpenAI, 2023; Anthropic, 2024). Despite this rapid progress, the major bottleneck in the real-world deployment of LLMs stems from their inference latency, due to the nature of auto-regressive decoding. Generating $n$ tokens requires $n$ sequential runs, making the process time-consuming and leading to under-utilizing available computation resources.

To address this challenge, recent works (Chen et al., 2023; Leviathan et al., 2023) have proposed *speculative decoding* to accelerate the inference. Speculative decoding first leverages a *draft model* to sample a bunch of tokens as candidates, which are later verified in parallel by the *target model*. If the verification of a token fails, its succeeding tokens must all be rejected to ensure output distribution is unbiased. Therefore, the performance of speculative decoding is strongly bounded by the *acceptance rate* of predicted tokens.

To this end, several methods have explored tree structures to enhance the acceptance rate, as illustrated in Figure 1. For instance, Sun et al. (2024) developed **SpecTr**, introducing DraftSelection algorithm to make draft model select multiple candidates while maintaining the same output distribution as the target model. Miao et al. (2023) created **SpecInfer**, which constructs token trees using small speculative models with learnable branch numbers of each layer. Similarly, Cai et al. (2024) proposed **Medusa**, which bases token tree construction directly on draft model probabilities, optimizing efficiency when the draft model closely approximates the target model. Meanwhile, Chen et al. (2024) introduced **Sequoia**, which estimates acceptance rates for candidate tokens and uses dynamic programming to optimize the token tree based on the estimated metric. However, a common limitation of these methods is their reliance on *fixed* patterns of tree construction, which

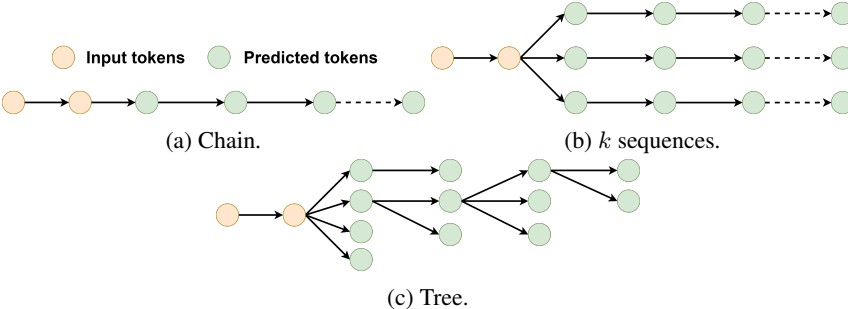

(a) Chain.

(b) $k$ sequences.

(c) Tree.

Figure 1: Different structures of predicted tokens. SpecTr is 1b structure, while Specinfer, Medusa and Sequoia are 1c structure.

can lead to suboptimal performance across diverse query distributions, resulting in a relatively low acceptance rate as tree size grows. This raises an important research question:

**RQ 1:** How can we find a *near-optimal* token tree structure for speculative decoding? To answer the research question, we will first establish the connection between acceptance rate and draft distribution through the following hypothesis.

**Hypothesis 1.** *Predicted tokens of higher draft probability statistically have a higher acceptance rate.*

Fortunately, this is further validated by our preliminary studies, as demonstrated in Figure 2. With the observation, we propose DYSPEC to *dynamically* expand the token tree based on draft distribution. DYSPEC employs a greedy search strategy to maximize the expected length of the predicted sequences. Compared with its fixed counterpart, the dynamic token tree yields a higher acceptance rate and speedup. We conduct benchmarking experiments on various datasets and different model scales, the experimental results demonstrate our proposed DYSPEC can efficiently improve the inference performance. Specifically, on the Llama2-70B model, DYSPEC achieves a $9.1\times$ throughput improvement and $9.4\times$ reduction in latency.

## 2 PRELIMINARY

**Speculative Decoding.** Chen et al. (2023) and Leviathan et al. (2023) proposed speculative decoding as a means to accelerate auto-regressive decoding. This approach samples generations from an efficient draft model as speculative prefixes and verifies these tokens in parallel using a slower target model. Through rejection sampling, it ensures that the outputs have the same distribution as those from the target model alone.

We denote the distribution of the draft model as $D[\cdot]$[1], and the target distribution as $T[\cdot]$. In speculative decoding, a token $x$ sampled from $D$ is accepted with a probability of $\min(1, \frac{T[x]}{D[x]})$. In case of rejection, another token $y$ will be sampled from a residual distribution $\mathtt{norm}(\mathtt{relu}(T - D))$ to adjust the output aligned with the target distribution.

**Tree Attention.** Transformer (Vaswani et al., 2017) models use the attention mechanism to aggregate sequential information. In implementation, the auto-regressive model uses an upper triangle mask to preserve causality. In the context of tree-based dependency, Liu et al. (2020) first proposed tree attention to represent the hierarchy as:

$$\mathtt{mask}(A)_{i,j} = \left\{ \begin{array}{ll} 1 & , \ i \text{ is ancestor of } j, \\ 0 & , \ \text{otherwise.} \end{array} \right.$$

In speculative decoding, tree attention has later been adopted by SpecInfer (Miao et al., 2023) and Medusa (Cai et al., 2024) for parallel verification.

---

[1]We use $D[\cdot]$ as an abbreviation of conditional probability $D(x_t|x_{<t})$, and similarly for $T[\cdot]$.

## 3 RELATED WORK

### 3.1 TREE-STRUCTURE SPECULATIVE DECODING

In this section we introduce the previous works of utilizing tree structure for speculative decoding in the LLMs' generating process.

**SpecTr.** Sun et al. (2024) proposed DraftSelection algorithm to make draft model select multiple candidates and maintain the same distribution of output as the target model. With the fix number of candidates $k$, they modeled an optimal transportation problem to find the best division factor $rho$ to maximize the acceptance rate, and proposed K-SEQ algorithm that extend k candidates to k sequences.

**SpecInfer.** Miao et al. (2023) proposed SpecInfer which leverages many small speculative model to construct the token tree, and make the branch number of each layer $k_i$ learnable.

**Medusa.** Cai et al. (2024) also introduce an optimized token tree construction. However, Medusa build the token tree directly based on the probability of draft model, instead of a mapping between sampling of draft model and sampling of target model. The second one make the speculative decoding maximize the efficiency if draft model are close to the target model.

**Sequoia.** Chen et al. (2024) estimates an acceptance rate vector for candidates by a few examples. Under the assumption that the expected acceptance rate of each candidate token is only related to the number of guesses it has been made, Sequoia use a dynamic programming method to get the optimized token tree.

**Eagle-2.** Li et al. (2024) proposed a speculative decoding method with dynamic predicted token tree. Eagle-2 is a self-speculative method that makes draft predictions based on the target model's features, rather than a much smaller draft model. Due to the strong drafting capability, self-speculative methods (Medusa, EAGLE, and EAGLE-2) can usually guess with higher accuracy under the same budget. Eagle-2 builds their draft trees with an expand-rerank procedure: first selects top-k tokens at each node, and prunes the candidate tree with draft probability. The main difference between Eagle-2 and DYSPEC is that DYSPEC does sampling at each node, and dynamically allocates the budget after the result of the sampling is determined. Eagle-2 greedily chooses the top-k draft token at each node and will accept the token if the target model generates the token in guessed tokens. EAGLE-2 cannot accept tokens with standard verification, i.e. only reject the draft with probability $1 - \frac{target}{draft}$ when $draft > target$, since draft tokens are predicted by selection rather than sampling. The problem here is that even in the case that draft probability is identical to target probability, the latter verification may yield a low acceptance rate. This building method is difficult to directly integrate into a standard verification framework, as the pruning operation can be seen as a rejection of certain sampled tokens, potentially affecting the generation probability distribution.

**ReDrafter.** Cheng et al. (2024) proposed a speculative decoding method with dynamic predicted token tree. ReDrafter uses beam-search-like method to extend the predicted token tree with maximum draft token probability. Since ReDrafter greedily choose the tokens in building stage instead of sampling, it cannot apply the standard verification.

**Dynamic Depth Decoding.** Brown et al. (2024) proposed a mechanism for tree-based speculative decoding methods to dynamically select the depth of the predicted token tree. This approach can be integrated with existing methods, many of which rely on a predetermined fixed depth. Furthermore, it can be combined with DYSPEC to optimize the threshold selection rather than the depth, thereby constructing the predicted token tree more efficiently and minimizing the number of draft model calls.

## 4 BRIDGING DRAFT DISTRIBUTION WITH ACCEPTANCE RATE

During verification, the acceptance probability of sampled token $x$ is given by $\min(1, \frac{T[x]}{D[x]})$. We now derive the connection between draft distribution and acceptance rate as follows.

Since the draft distribution acts as the approximation of the target distribution, the two distributions should not be too "far" away. Without loss of generality, we assume that the KL divergence of $D$

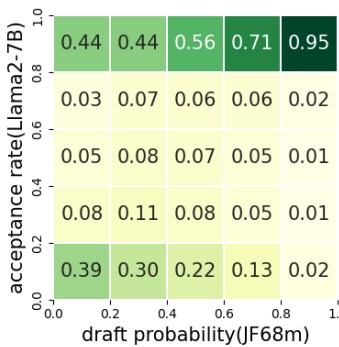 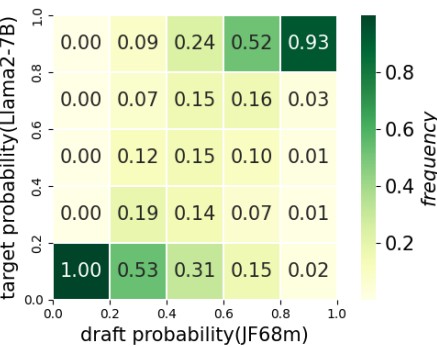

Figure 2: Connection between acceptance rate/target distribution and draft distribution on CNN DailyMail.The density of each block is normalized by column.

from $T$ is constrained by constant $c$, i.e.,

$$D_{\mathrm{KL}}(D \parallel T) = \sum D[x] \log \frac{D[x]}{T[x]} \leq c. \tag{1}$$

To satisfy the constraint, $T[\cdot]$ should not diverge much from $D[\cdot]$. Nevertheless, for a token $x$ with large draft probability $D[x]$, $\frac{T[x]}{D[x]}$ cannot be too small, as it would contribute significantly to $D_{\mathrm{KL}}$. On the other hand, tokens with small $D[x]$ have less impact to $D_{\mathrm{KL}}$, allowing for greater variation. The above analysis implies that **predicted tokens of higher draft probability statistically have a higher target probability and acceptance rate**.

We further validate our hypothesis through preliminary experiments. As shown in Figure 2 (right), the draft distribution shows a strong correlation with the target distribution in real-world scenarios. More importantly, Figure 2 (left) demonstrates that the distributions of acceptance rate, under the same draft probability, resemble binomial distributions. As draft probability grows larger, predicted tokens are more likely to be accepted. These observations provide strong empirical support for our previous claim. It also inspires us to design a dynamic token tree construction algorithm to explore more on sub-trees of higher draft probability, since they are more likely to be accepted in later verification.

## 5 METHOD

Under a fixed speculative budget $b$ (i.e. the number of tokens for each verification), the optimal token tree yields the highest acceptance rate. In practice, finding the optimal tree is unfeasible, since the target distribution is unknown before verification. Nevertheless, given Hypothesis 1, we can transform the original problem into the following problems.

### 5.1 DYNAMIC TOKEN TREE CONSTRUCTION

Given the speculative token tree, the way we sampling this tree, the draft model output distribution, and correspond target model output distribution, we can get the expectation of the total number of Speculative decoding verification. Considering each node $t_i$ in speculative token tree independently, we denote its draft distribution as $p_d[i, \cdot]$, and the relevant target distribution as $p_t[i, \cdot]$.

Assume that node $t_i$ have ancestors $a_1, ..., a_i$, and previous sibling node $s_1, ..., s_j$, then the probability we verify the node $t_i$ can be represent as $\prod_i P[accept a_i] \times \prod_j P[reject s_j]$.

In Speculative Decoding, the probability we accept token $x$ with draft probability $p_d[x]$ and target probability $p_t[x]$, is $\min(1, \frac{p_t[x]}{p_d[x]})$, denote as $SD[x]$. So the probability we take verification on node $t_i$ is $\prod_i SD[a_i] \times \prod_j (1 - SD[s_j])$. Then the contribution of node $t_i$ to expectation of total accepted token number is $\prod_i SD[a_i] \times \prod_j (1 - SD[s_j]) \times SD[t_i]$.

The total expectation of accepted token number of this speculative token tree is

$$\sum_u \prod_i SD[a_{i,t_u}] \times \prod_j (1 - SD[s_{j,t_u}]) \times SD[t_u] \tag{2}$$

With expected acceptance rate, we can construct the optimal speculative token tree. However, there are still two problems:

1. When we generate speculative token tree, we cannot know the target probability to get $SD[\cdot]$.

2. The draft token $t_i$ is sampled from draft output distribution, we could only decide how many sampling we take, instead of which token to take. Otherwise the take action we made will infect the probability we keep tokens in speculative token tree.

To solve problem 1, we note that the acceptance rate is positive-related to draft output distribution. Given Hypothesis 1, we use draft model output distribution to estimate the acceptance rate $SD[t_i] \approx p_d[t_i]$.

To solve problem 2, we only use these estimated values to decide if we will make the sampling. For given intermediate token tree status, we can detect all expandable tree nodes, and pick the expandable tree node with maximum estimated value. Repeat this action until we reach the max tree size, DYSPEC will generate the optimal speculative token tree. The proof of optimality is provided in Appendix D.

Now we can get the algorithm to generate the optimal speculative token tree.

## 5.2 ALGORITHM

Unlike some speculative decoding methods, DYSPEC determines the number of samples to take only when a token is accepted by the target model (or the verification method). This decision is based on the verification results of the previous tokens (ancestor nodes in the predicted token tree) and the previous sampling results from the same node. There are two kinds of operation of the number of samples: 1. from 0 to 1(expand a node with no leaf no, the first sampling). 2. from $x$ to $x+1$(failed on the $x$-th sampling, take the $x+1$ sampling).

Given the prompt, DYSPEC can get the logits of the last token, which is the root of the speculative token tree. Suppose we have already constructed a partial speculative token tree as Figure 3. There are two ways to expand a node:

1. Any token without a leaf node can undergo the first sampling.

2. Nodes marked with "–/–" indicate that we have already performed several samplings at the same position and have obtained an estimated value for the next sampling at this position (on the arrow line). The "–/–" node corresponds to the result of the next sampling.

We refer to these two types of nodes as expandable nodes in the current state.

DYSPEC use a heap to maintain all the expandable tokens by their estimated values, that we can get the node with maximum estimated value in $O(logN)$ time. After we make the next sampling represented by the top node of the heap. Upon determining the result of the sampling, we then update the state of the current token tree using the obtained token and its corresponding estimated value. This process generates two new expandable nodes:

1. When the current node is *rejected*, the next sampling at the same position, with the corresponding estimated value being the probability of this sampling failure multiplied by the expected acceptance rate of the next sampling itself.

2. When the current node is *accepted*, proceeding with subsequent sampling, with the corresponding estimated value being the probability of this sampling success multiplied by the expected acceptance rate of the next sampling itself.

Thus, we have successfully expanded the token tree by one node. This process is repeated until the predetermined budget is reached. The pseudo-code is presented in Algorithm 1.

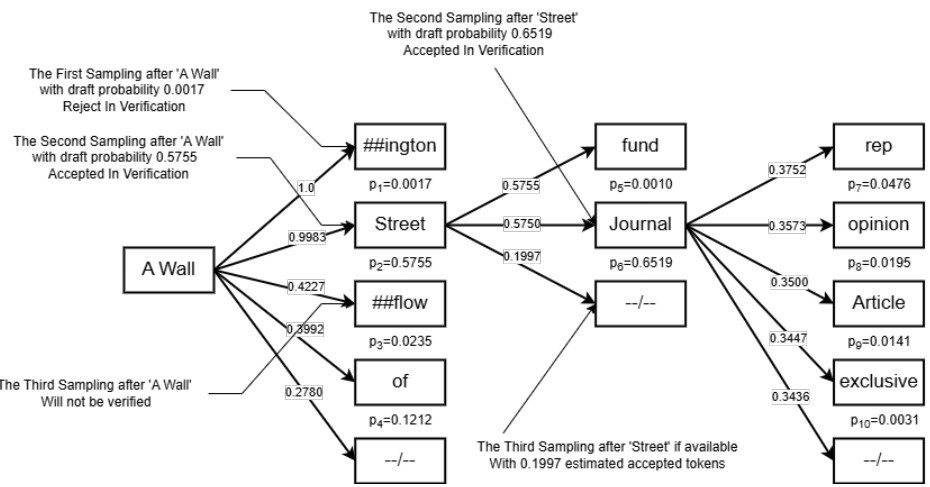

Figure 3: An example of the predicted token tree.

---

**Algorithm 1:** Speculative token tree construction algorithm with fixed number

**Input** : Prefix $x_0$, draft model $D_\Theta(\cdot|x)$, and an upper bound of guess tokens number $m$.
**Output:** generated token tree $Tr$.

1 Initialize a heap $H$, Heap Element consists of tree information $\texttt{TreeInfo}_i$, residual
   distribution $R_i$, estimate acceptance rate $v$.
2 $R \leftarrow D_\Theta(\cdot|x_0), v \leftarrow 1, \texttt{TreeInfo} \leftarrow \ldots$
3 $H.push(R, v, \texttt{TreeInfo})$;
4 **while** $Tr.size < m$ **do**
5    |  $R, v, \texttt{TreeInfo} \leftarrow H.pop()$;
6    |  $\texttt{NewNodeInfo} \leftarrow \text{Tr.add}(\texttt{TreeInfo}, \text{y})$;
7    |  sample $y \sim R$ ;
8    |  $v_0 = v \times R[y]$ ;
9    |  $v_1 = v \times (1 - R[y])$ ;
10   |  $R[y] \leftarrow 0$;
11   |  $R \leftarrow norm(R)$;
12   |  $H.push(R, v_1, \texttt{TreeInfo})$ ;         /* expand neighbor node */
13   |  get $x_i$ from TreeInfo and y;
14   |  $d_i \leftarrow D_\Theta(\cdot|x_i)$;
15   |  $H.push(d_i, v_0, \texttt{NewNodeInfo})$ ;      /* expand child node */
16 **end**

---

### 5.3 ANALYZE OVERHEAD

Assume the speculative token tree size is $N$, depth is $D$. Greedy expand method will generate the optimal token tree one by one. For each token, greedy expand method choose the expandable token with maximum estimated valueand then make a sampling to generate the next token, then update the token tree.

To quickly choose the expandable token with maximum estimated value, we can use heap to maintain all expand-able tokens' estimated value, which introduce $O(logN)$ time complexity to maintain the token tree and related auxiliary structures. The total time complexity of token tree construction is $O(NlogN)$.

Although one step inference's time consume of draft model is usually much lower than target model, it is still non negligible. Denote draft model inference time as $T_d$, target model inference time as $T_t$, the total time of one step of greedy expand method is

$$O(NlogN + T_t + NT_d) \tag{3}$$

With accepted token number $e$, the latency of generate one token can be represent as $O((N log N + T_t + NT_d)/e)$.

In the implementation, the time complexity of constructing a token tree for a single operation is $O(vocab\_size)$, due to the sampling and updating of the residual distribution. Typically, the inference of a draft model involves higher time complexity. However, model inference benefits from regular computational workloads and can be efficiently accelerated by GPUs, whereas the complex logical operations involved in token tree construction suffer from low efficiency when implemented in Python. To mitigate this overhead, we implemented the token tree construction in C++, making it negligible compared to the inference times of both the target and draft models.

Even if we disregard the overhead associated with constructing the token tree, accelerating the target model still requires us to achieve a speedup factor of approximately $k \approx 1/e + \frac{NT_d}{eT_t}$, where $1/k$ represents the acceleration rate. As the number of tokens $N$ increases, the term $N/e$ grows significantly. For instance, with $N = 64$, $N/e$ typically exceeds 10 , and for $N = 768$, $N/e$ can surpass 70. This rapid growth severely limits the potential for acceleration by simply increasing the size of the token tree.

To address this limitation, we need to develop a more efficient method for generating draft tokens. It's important to note that the token tree structure will branch out significantly after a few steps, resulting in a relatively shallow depth. If we can generate draft tokens layer by layer, the latency for generating one token can be represented as $O((N log N + T_t + DT_d)/e)$, where the time cost of one step can be considered constant for an appropriate input size. For $N = 64$, $D$ is typically less than 10, and for $N = 768$, $D$ is usually less than 30.

However, the greedy expansion method struggles to align with layer-by-layer generation because, without revealing the estimated values of all tokens, it is challenging to determine how many tokens should be included in the shadow layers.

## 5.4 Construct Token Tree with Threshold

To accelerate inference, we must reduce the number of draft generations. In the greedy expansion method, we select the token with the highest estimated value at each step, and this value monotonically decreases with each selection. Once the token tree construction is complete, all tokens with an estimated value greater than a certain threshold $C$ are chosen, while those with lower values are discarded. If we could determine this threshold $c$ at the outset, it would be possible to construct the optimal speculative token tree layer-by-layer. In practice, we can choose an appropriate threshold $C$ (typically around $1/n$) and relax the constraint on $N$. This adjustment has a minimal impact on the number of accepted tokens but significantly improves latency. The pseudo-code is provided in Appendix A.2.

## 6 Empirical Results

### 6.1 Setup

We implement DYSPEC using Llama models. We employs JackFram/Llama68m (JF68m) and Llama2-7B as the draft model, and Llama2-7B, Llama2-13B, Llama2-70B (Touvron et al., 2023) as the target models. We conduct evaluations on various datasets with varying sizes and characteristics, including C4(en) (Raffel et al., 2020), OpenWebText (Gokaslan & Cohen, 2019) and CNN DailyMail (Nallapati et al., 2016).

For a fair comparison, we follow the setting in Sequoia (Chen et al., 2024), using the first 128 tokens as the fixed prompt and generating 128 tokens as completion. We evaluate our method with different target temperatures and set the draft temperature to 0.6. All experiments are conducted on a computation node with one NVIDIA A100 40GB GPU and 32 CPU cores.

### 6.2 Overhead of tree construction

As analyzed in the Section 5.3, the construction of the token tree introduces complex logic, which is inefficient in Python despite its time complexity of $O(N log N vocab\_size)$. To address this, we

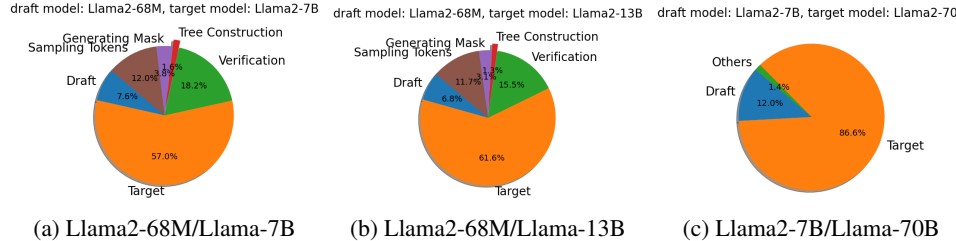

(a) Llama2-68M/Llama-7B   (b) Llama2-68M/Llama-13B   (c) Llama2-7B/Llama-70B

Figure 4: The execution times of different components during the inference process.

implemented the construction in C++, making the construction time negligible. The profiling results are shown in Figure 4. The additional overhead introduced by DYSPEC is the *Tree Construction*, which accounts for less than two percent of the total execution time in the Llama2-68M/Llama2-7B and Llama2-68M/Llama2-13B pairs. In the Llama2-7B/Llama2-70B pair with CPU-offloading, all components except draft and target model inference cost less than two percent of the total execution time.

The generation of masks, sampling tokens, and verification consume significant time under both the Llama2-68M/Llama2-7B and Llama2-68M/Llama2-13B settings. These three components represent the common overhead of all speculative decoding methods, with the primary time spent on waiting for the completion of model execution via CUDA synchronization. In the Llama2-7B/Llama2-70B setting, CPU-offloading and waiting for model execution results overlap, which is why they are not reflected in the profiling results.

## 6.3 EFFECTIVENESS OF DYNAMIC TOKEN TREE

Table 1 presents the experimental results, detailing the number of tokens accepted and the latency per token in seconds, when using JF68M as the draft model and Llama2-7B as the target model. Similarly, Table 2 shows the corresponding results for the scenario in which JF68M serves as the draft model and Llama2-13B as the target model. In both cases, the maximum draft token tree size is set to 64. For the draft model, DYSPEC leverages the CUDA graph to capture 129 different input lengths ranging from 128 to 258, thus accelerating inference, much like Sequoia does.

The results indicate that DYSPEC consistently outperforms Sequoia and Specinfer in various data distributions and generation temperatures, leading to a higher number of accepted tokens in each decoding step. The values in the table represent the average time taken to generate a single token in seconds, with the number of tokens accepted by the target model during a single validation in parentheses.

For larger target models such as Llama2-70B, we employ CPU offloading due to GPU memory constraints. We selected Llama2-7B as the draft model. Despite the time consumed for data synchronization between the CPU and GPU, the inference time for the CPU-offloaded model, with a naive implementation, is approximately 15 seconds per step. By incorporating some overlapping tricks for weight loading (adapted from Sequoia), the inference time is still around 5 seconds per step. In contrast, Llama2-7B requires only about 25 milliseconds per step, resulting in a $T_t/T_d$ ratio of approximately $2 \times 10^3$. Note that DYSPEC did not employ CUDA Graph in this scenario due to the significant GPU memory overhead associated with capturing sequences of varying lengths. With 129 distinct sequence lengths and the memory-intensive nature of the draft model Llama2-7B, this approach would be prohibitively resource-demanding.

In this scenario, the acceleration rate is roughly equivalent to the number of tokens accepted per target model step. Set the maximum draft token tree size to 64, DYSPEC achieves up to a 9.1x improvement in throughput and a 9.4x reduction in latency compared to auto-regressive generation, while also outperforming state-of-the-art methods in consistency, as demonstrated in Table 3.

Table 1: latency per token. The draft model is JF68m and the target model is Llama2-7B. Guess length is 64.

| Dataset | Temp | Ours | Sequoia | Specinfer |
|---|---|---|---|---|
| C4 | 0 | 3.15×(5.25) | 2.64×(4.99) | 1.79×(3.32) |
| C4 | 0.6 | 2.20×(3.71) | 1.85×(3.45) | 1.80×(3.44) |
| OWT | 0 | 2.28×(3.79) | 2.19×(3.81) | 1.47×(2.54) |
| OWT | 0.6 | 2.40×(3.07) | 2.18×(3.04) | 2.09×(2.97) |
| CNN | 0 | 2.42×(3.97) | 2.40×(4.04) | 1.53×(2.58) |
| CNN | 0.6 | 2.09×(3.18) | 1.99×(3.22) | 1.80×(3.06) |
| GSM8k | 0 | 3.93×(6.86) | 2.79×(4.92) | 2.01×(3.47) |
| GSM8k | 0.6 | 2.44×(4.31) | 2.20×(3.55) | 1.65×(3.03) |
| MT-Bench | 0 | 2.59×(4.02) | 2.35×(3.55) | 1.68×(2.70) |
| MT-Bench | 0.6 | 2.15×(3.62) | 2.11×(3.18) | 1.50×(2.71) |

Table 2: latency per token. The draft model is JF68m and the target model is Llama2-13B. Guess length is 64.

| Dataset | Temp | Ours | Sequoia | Specinfer |
|---|---|---|---|---|
| C4 | 0 | 3.13×(4.98) | 2.66×(4.35) | 1.97×(3.14) |
| C4 | 0.6 | 2.26×(3.62) | 1.88×(3.15) | 1.85×(3.15) |
| OWT | 0 | 2.45×(3.59) | 2.33×(3.44) | 1.67×(2.44) |
| OWT | 0.6 | 1.96×(3.02) | 1.78×(2.80) | 1.71×(2.75) |
| CNN | 0 | 2.56×(3.82) | 2.45×(3.67) | 1.69×(2.52) |
| CNN | 0.6 | 2.03×(3.11) | 1.84×(2.91) | 1.78×(2.84) |
| GSM8k | 0 | 3.17×(5.29) | 2.21×(3.92) | 1.74×(2.98) |
| GSM8k | 0.6 | 2.49×(4.17) | 2.10×(3.39) | 1.51×(2.72) |
| MT-Bench | 0 | 2.19×(3.72) | 2.19×(3.46) | 2.15×(2.86) |
| MT-Bench | 0.6 | 2.25×(3.62) | 1.93×(3.11) | 1.40×(2.84) |

Table 3: latency per token. The draft model is Llama2-7B and the target model is Llama2-70B. Guess length is 64.

| Dataset | Temp | Ours | Sequoia | Specinfer |
|---------|------|------|---------|-----------|
| C4 | 0 | 9.42×(9.10) | 6.29×(6.08) | 4.89×(4.67) |
| C4 | 0.6 | 6.77×(6.21) | 5.66×(5.72) | 5.76×(5.75) |
| OWT | 0 | 7.07×(7.23) | 6.02×(6.41) | 5.07×(4.88) |
| OWT | 0.6 | 6.05×(6.77) | 5.63×(6.07) | 5.42×(5.46) |
| CNN | 0 | 6.50×(6.93) | 5.85×(6.42) | 4.80×(4.83) |
| CNN | 0.6 | 5.94×(6.95) | 5.71×(6.07) | 5.70×(5.75) |
| GSM8k | 0 | 10.56×(12.39) | 7.31×(7.62) | 5.22×(5.34) |
| GSM8k | 0.6 | 7.57×(8.14) | 6.62×(6.89) | 5.75×(5.89) |
| MT-Bench | 0 | 9.95×(11.25) | 6.96×(7.46) | 4.75×(4.85) |
| MT-Bench | 0.6 | 8.47×(10.11) | 6.96×(7.46) | 5.52×(5.67) |

## 7 CONCLUSION

We introduce DYSPEC, a faster speculative decoding algorithm that incorporates a dynamic token tree structure for sampling. Based on the connection between draft probability and acceptance rate, we apply a greedy strategy to dynamically expand the token tree to maximize the expected length of predicted generations. Empirical results reveal the efficacy and scalability of DYSPEC by consistent improvements in acceptance rate across various datasets and generation temperatures. Specifically, on the Llama2-70B model with temperature=0, DYSPEC achieves a 9.1× throughput improvement and 9.4× reduction in latency.

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

## A    TOKEN TREE CONSTRUCTION ALGORITHM

We present the details of our token tree construction algorithms and the corresponding verification method to ensure that the output probability distribution is consistent with the target model.

### A.1    TOKEN TREE CONSTRUCTION ALGORITHM WITH FIXED SIZE

We demonstrate the proposed token tree construction algorithm with fixed size in Algorithm 1.

The optimal predicted token tree can be generated by greedily expanding the leaf node with the highest expectation. This method can be implemented using priority queues, similar to REST He et al. (2023).

Assume that we have a partial token tree. Then we use a heap to maintain all extendable nodes (leaf nodes or the last predicted node of its parent). Each time we extend the extendable node with the highest estimated acceptance rate. After adding one node to token tree, there are two more extendable node. One is its first child(the first prediction following this token). This prediction will only occur if the current node is received, so its estimated acceptance rate is `previous_rate` $\times p$, where $p$ is the estimated acceptance rate of current token. The other extendable node is its next neighbor(the next prediction of the same previous tokens). This prediction will only occur if the current node is rejected, so its estimated acceptance rate is `previous_rate` $\times (1 - p)$.

The algorithm starts with a single root node, which represents the input prefix. Then repeat the aforementioned process $m$ times. The estimated acceptance rate of the node can be expressed as the product of its all ancestor nodes' probability multiply the probability that all its previous predictions failed under the same prefix tokens. The new extendable nodes (i.e., $v_0$ and $v_1$ in Algorithm 1) should have the lower estimated acceptance rate than previous predicted tokens. It means that we generated tokens with decreasing acceptance rate and the residual nodes remain in heap or are not extendable have lower acceptance rate than any generated tokens, which means that we get the optimal token tree.

Note that the estimated acceptance rate is independent of its actual token, because we made this prediction before we know what the token is. If what this token is affects whether or not we keep the sample in draft token tree, then the final result will be biased.

Algorithm 1 will call draft model $m$ times, which is inefficient for large $m$. An alternative way is generating predicted tokens layer by layer. To do this, we can relax the fixed $m$ limitation to an appropriate threshold. Algorithm 1 will greedily generate the first $m$ nodes with largest estimated acceptance rate. If we set the threshold to be the same as the acceptance rate of the last token, we will exactly get the same result as the previous algorithm. And it will only call the draft model *layer number* times.

### A.2    TOKEN TREE CONSTRUCTION ALGORITHM WITH THRESHOLD

We present our token tree construction algorithm with threshold in Algorithm 2. The different between Algorithm 1 and Algorithm 2 is that we extend all nodes with estimated acceptance rate above the threshold.

### A.3    VERIFICATION

After the process of token tree, we need a corresponding verification method to ensure that the output probability distribution is consistent with the target model. Our method can be seen as the method dynamically choose the branch number of each token. So the verification method is similar to SpecInfer (Miao et al., 2023) and Sequoia (Chen et al., 2024). We present our verification algorithm in Algorithm 3.

The major difference between Sequoia and ours is that we directly return when the distribution of draft output become all zeros. In that case the estimated acceptance rate in our method is 0 and will never be extended.

---

**Algorithm 2:** Token tree construction algorithm with threshold

**Input** : Prefix $x_0$, draft model $D_\Theta(\cdot|x)$, and a threshold $t$.
**Output:** generated token tree $Tr$.

1   $R \leftarrow D_\Theta(\cdot|x_0), v \leftarrow 1, \texttt{TreeInfo} \leftarrow \ldots$
2   LeafNodes $\leftarrow$ root;
3   **while** $\textit{LeafNodes} \neq \emptyset$ **do**
4      NewLeafNodes $\leftarrow \emptyset$ ;
5      **foreach** $node_i \in \textit{LeafNodes}$ **do**
6         get input $x_i$ from $\texttt{node}_i$;
7         $d_i \leftarrow D_\Theta(\cdot|x_i)$;
8         get estimate acceptance rate $v_i$ from $\texttt{node}_i$ ;
9         **while** $v_i < t$ **do**
10            sample $y \sim d_i$ ;
11            NewNode $\leftarrow Tr.\text{add}(node_i, y)$ ;
12            NewLeafNodes.append(NewNode, $v_i * d_i[y]$) ;   /* expand child node */
13            $v_i = v_i * (1 - d_i[y])$;
14            $d_i[y] = 0$;
15            $d_i \leftarrow norm(d_i)$;
16         **end**
17      **end**
18      LeafNodes $\leftarrow$ NewLeafNodes ;
19 **end**

---

# B   ADDITIONAL EXPERIMENTS

For all experiments, we selected 1000 pieces of data from each dataset to conduct the experiment. For CNN daily we used test splits. For openwebtext we used train split. For C4 we used en splits. All the results were the result of a single run.

## B.1   DYSPEC WITH LARGE TOKEN TREE SIZE

Under CPU-offloading setting, the target model inference is extremely larger than the draft model. For Llama2-70B as target and llama2-7b as draft on A100 40G, target model inference time is 2000 $\times$ larger than draft model, which gives us the opportunity to construct a larger token tree. Following Sequoia's setting, we also make the guess token tree size up to 768. The result shows that our method can achieve a higher accepted token per step, and lower latency per token than SOTA at 0 target temperature.

On higher temperatures, DYSPEC demonstrates superior performance compared to Specinfer, but it does not surpass Sequoia. This is due to efficiency constraints that prevent us from implementing the full version of DYSPEC's greedy method. Instead, we must employ a threshold to construct the token tree layer by layer. The exact threshold varies over time, which limits our ability to fully utilize the 768-token budget. For instance, at a target temperature of 0.6 on the OpenWebText dataset, with a maximum tree size set to 768 and a threshold of 0.001, the average tree size is 551.79. Figure 5 illustrates the token tree size at each step alongside the number of accepted tokens.

To maximize the potential of DYSPEC's greedy expansion method, we need to develop mechanisms for dynamically adjusting the threshold or create an alternative algorithm that eliminates the draft model inference overhead while preserving the token-by-token expansion mechanism.

# C   BLOCK-SPARSITY FRIENDLY TOKEN ORDER

The special sparsity in tree attention brings opportunity to further optimize the attention operation. Since modern attention libraries (e.g. FLASHATTENTION) compute block by block, different token permutations can have distinct computation workloads. To find the optimal token order, we formalize the optimization problem as below:

**Algorithm 3:** Verify Algorithm

**Input**  : draft model distribution $Draft(\cdot)$, target model distribution $Target(\cdot)$, speculated
         token tree Tr.

**Output:** Accepted token sequence $A$.

1 CurrentNode ← Tr.root;
2 $A \leftarrow \emptyset$;
3 **while** *CurrentNode.branches $\neq \emptyset$* **do**
4     $D \leftarrow Draft(\texttt{CurrentNode}, \cdot)$;
5     $T \leftarrow Target(\texttt{CurrentNode}, \cdot)$;
6     $R \leftarrow T$;
7     **for** *$node_i \in CurrentNode.branches$* **do**
8        get token $y$ from node_i ;
9        sample $c \sim N(0,1)$;
10        **if** $c \leq \frac{R[y]}{D[y]}$ **then**
11           A.append(y);
12           CurrentNode ← node_i;
13           break;
14        **else**
15           $R \leftarrow norm(max(R - D, 0))$;
16           $D[y] \leftarrow 0$;
17           **if** *$D$ is all 0* **then**
18              break;
19           **end**
20           $D \leftarrow norm(D)$;
21        **end**
22     **end**
23     **if** *CurrentNode isn't updated* **then**
24        sample $y \sim R$ ;
25        A.append(y);
26        break;
27     **end**
28 **end**

Table 4: Latency per token(accepted token per step). The draft model is Llama2-7B and the target model is Llama2-70B. Guess length is 768.

| Dataset | Temp | Ours | Sequoia | Specinfer | Baseline |
|---------|------|------|---------|-----------|----------|
| C4  | 0   | 0.42412(16.04) | 0.62841(9.40) | 0.86(8.66)* | 5.59650 |
| C4  | 0.6 | 0.88494(7.14)  | 0.66293(8.96) | 1.09(6.93)* | 5.34781 |
| OWT | 0   | 0.54885(11.79) | 0.62979(9.81) | 1.02(7.36)* | 5.52462 |
| OWT | 0.6 | 0.81002(7.66)  | 0.65147(9.12) | 1.21(6.18)* | 5.30340 |
| CNN | 0   | 0.54739(11.46) | 0.60206(9,54) | 0.95(7.87)* | 5.31049 |
| CNN | 0.6 | 0.87648(7.02)  | 0.65835(8.80) | 1.02(6.24)* | 5.29280 |

This data is sourced from Chen et al. (2024).

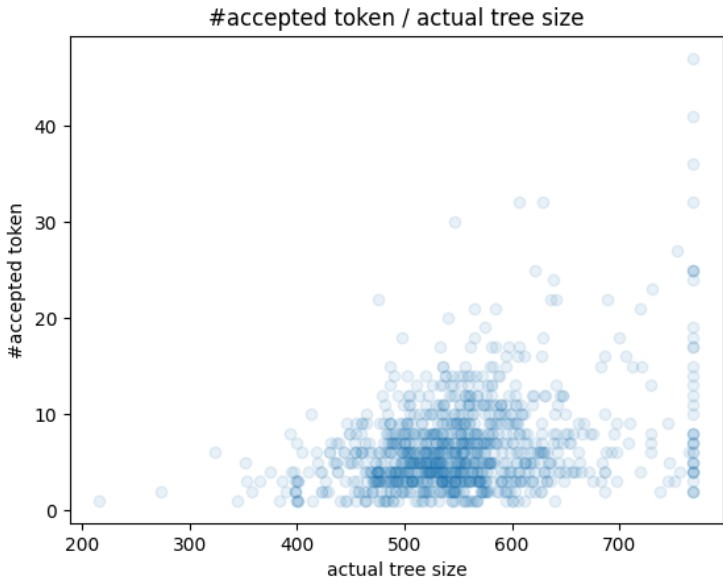

Figure 5: Token Tree size with accepted token number each step.

**Definition 1** (Block-Sparsity Friendly Token Order). *Given a tree $\mathcal{T}$ with size $n$ and computation block size $b$, find a permutation $\mathcal{P}$, s.t. the attention mask of tree $\mathcal{P}(\mathcal{T})$ has the minimal number of non-zero blocks.*

Exhaustively searching through all permutations is computationally prohibitive. A near-optimal solution to this problem is heavy path decomposition (HPD) (Sleator & Tarjan, 1981), which traverses nodes in descending order of their subtree sizes. This approach is effective because it groups nodes along longer paths into the same blocks whenever possible, while the long path contribute a lot to the total number of blocks in the tree attention mask ($O(L^2)$ blocks for path with length $L$). Given the way DYSPEC constructs the speculative token tree, previous sibling nodes are often allocated more budget to constrain their subtrees. Consequently, the depth-first search (DFS) order closely approximates the HPD order. DYSPEC leverages DFS to rearrange node indices, thereby reducing the number of non-zero blocks in the attention mask. As illustrated in Figure 6 and Figure 7, DFS order is typically more conducive to block sparsity.

C.1 EFFICIENCY OF OPTIMIZED TREE ATTENTION

For different tasks, there exist diverse patterns of attention masks. In response to the block sparsity of these masks, numerous implementations of attention operators based on FlashAttention have been developed, However, those methods are not well-suited to support arbitrary patterns of attention masks. XFormers (Lefaudeux et al., 2022) and DeepSpeed (Rasley et al., 2020) have no specific API for arbitrary custom mask. Recently, PyTorch (Paszke et al., 2017) introduces FlexAttention, which optimizes for arbitrary attention masks. However, to fully leverage its optimization, we must compile the kernel for different masks, which is not suitable for our target scenario of tree-based speculative decoding, where the tree attention mask changes with each iteration.

We have implemented a version of FlashAttention that supports custom masks, enabling the efficient handling of empty blocks in Triton (Tillet et al., 2019). Our experiments with a random tree attention mask demonstrate that DYSPEC Tree Reordering can reduce the number of attention mask blocks by up to $5.9\times$, and the attention operation can run up to $2.1 \times$ faster, as detailed in Table 5.

In the experiment, we set Q, K, V as shape (batch=1, head_num=64, seqlen, head_dim=128), where head_num=64 and head_dim=128 is the parameter used by Llama2-70B. The block size is 32, which is usually used in attention kernel according to limited shared memory size, and it can also provide considerable block sparsity. The seqlen is varies from 256 to 2048. We also compared our custom kernel with Manual Attention and Xformer, which demonstrates that our implementation kernel is on

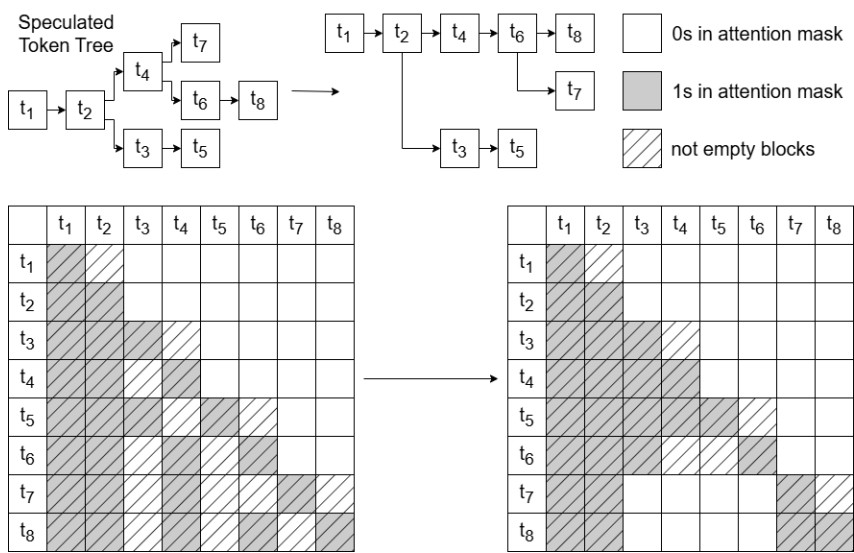

Figure 6: Comparing DFS order with original order.

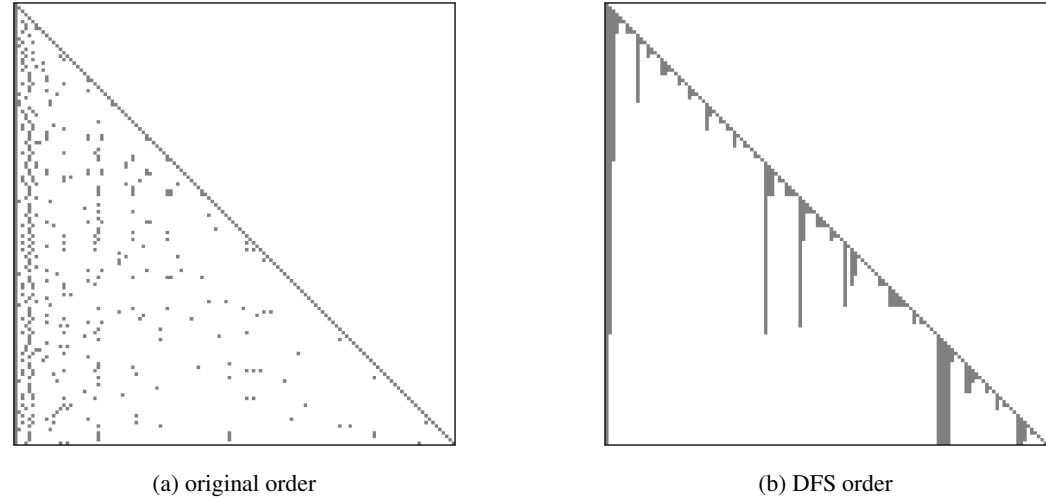

(a) original order    (b) DFS order

Figure 7: Tree attention mask of predicted token tree in different order.

par with the on-shelf kernel in terms of performance. And the negligible performance improvement of this kernel demonstrates that the performance enhancement of our method is entirely attributable to the reduction in the number of blocks.

In our experiment, we configured Q, K, and V with the shape (batch=1, head_num=64, seqlen, head_dim=128), aligning with the parameters used by Llama2-70B, where head_num=64 and head_dim=128. The block size was set to 32, a common choice in attention kernels due to the constraints of shared memory size, which also facilitates significant block sparsity. The sequence length (seqlen) varied from 256 to 2048. We benchmarked our custom kernel against Manual Attention and Xformers, revealing that our implementation performs comparably to existing kernels. The marginal performance improvement observed in those kernels underscores that the enhanced performance of our method is entirely due to the reduction in the number of blocks.

However, this improvement is not significant in end-to-end situation. These are two problems:

1. The improvement is only significant with large context length, where extremely large sizes will result in diminishing marginal benefits of increasing size on the acceptance rate of speculative de-

Table 5: Efficiency of Optimized Tree Attention with random tree structure.

| Tree Size | Reorder | custom kernel | Manual Attn | Xformer | Block Count |
|---|---|---|---|---|---|
| 256 | False | 0.07548 | 0.14089 | 0.17559 | 36 |
| 256 | True | 0.05406 | 0.14124 | 0.16721 | 22.5 |
| 512 | False | 0.21317 | 0.56264 | 0.15985 | 135.5 |
| 512 | True | 0.11364 | 0.55965 | 0.17285 | 52.8 |
| 1024 | False | 0.63368 | 2.08612 | 0.49049 | 490.2 |
| 1024 | True | 0.31801 | 2.08142 | 0.48922 | 119.3 |
| 2048 | False | 2.27148 | 9.20739 | 1.87807 | 1654.5 |
| 2048 | True | 1.02645 | 9.13469 | 1.87753 | 278.7 |

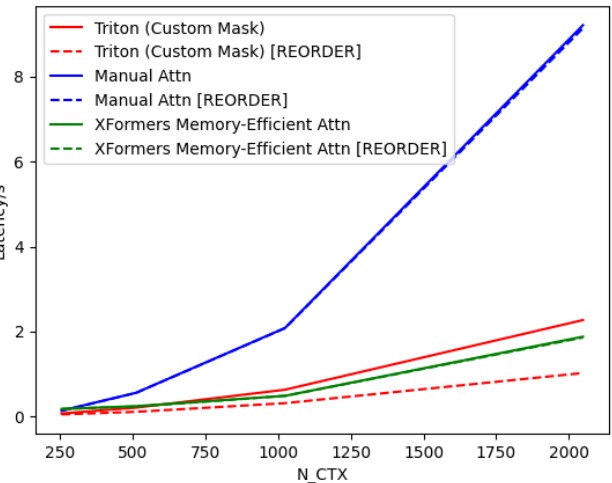

Figure 8: Efficiency of Optimized Tree Attention with random tree structure.

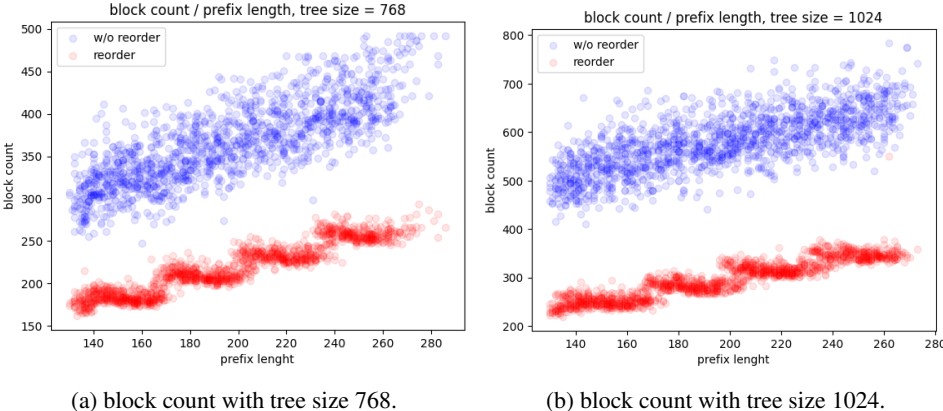

(a) block count with tree size 768.  (b) block count with tree size 1024.

Figure 9: Block Count with tree attention mask with/without tree reorder, with different prefix length.

coding. Despite the decline in acceptance rate as tree size increases, the ratio of inference speeds between the target model and the draft model itself limits the size of the tree.

Using large model like Llama2-70B with CPU-offloading will the ratio of inference speeds between the target model and the draft model, however, there is a new problem that under this setting, the most time cost operation is moving weight between CPU and GPU, and the attention operation only contribute a little in end–to-end latency.

2. The prompt is included in attention mask. As the context becomes longer, the majority of the attention calculations involve interactions between the newly added tokens and the existing context tokens. Consequently, the influence of the tree structure diminishes.

Figure 9 illustrates the block count on a real workload tree attention mask with varying prefix lengths. Specifically, for a tree size of 768, the block count with reordering is 218.31, compared to 366.12 with the original order. Similarly, for a tree size of 1024, the block count with reordering is 295.59, while it is 580.07 with the original order.

Only when these two issues are resolved can reordering effectively accelerate the end-to-end latency of tree-based speculative decoding. The first issue requires a more advanced speculative decoding method capable of handling extremely large tree sizes. The second issue likely necessitates optimizing the attention computation between the prompt sequence and new tokens, thereby shifting the bottleneck to the tree attention mask itself.

## D PROVE

The goal is to maximize the expected total acceptance tokens, denoted as $T = \sum_i p_i$, where $p_i$ represents the expected acceptance rate of token $t_i$ within the predicted token tree.

Given the assumptions that (1) the probability of a token appearing in the draft model outputs, denoted as $draft_i$, can approximate its acceptance rate, and (2) the acceptance rate of a token is independent of its preceding tokens, we can express the expected acceptance rate $p_i$ as:

$$p_i \approx P[Path_i]draft_i \tag{4}$$

Where $P[Path_i]$ represents the probability of accepting all the ancestor tokens of $t_i$ in the predicted token tree.

For multi-branch tokens under the same ancestor path, the acceptance of subsequent tokens is depends on the rejection of preceding sibling tokens. Assuming all ancestor tokens along the path have been accepted, the probability of verifying token $t_k$ can be expressed as:

$$P[verify_i|Path_i] = \prod_{j<k}(1 - draft_j) \tag{5}$$

Where $t_{j<k}$ denote $t_k$'s previous sibling tokens.

Put all three component together, we have

$$p_i = P[Path_i] \times \prod_j j < k(1 - draft_j) \times draft_k \qquad (6)$$

Although we have a method to estimate the expected acceptance token number, there are still challenges in finding the optimal structure for speculative decoding. The expectation can only be known after we have completed the sampling process. After sampling, the predicted token tree must be updated, otherwise some tokens with low acceptance rates will be pre-pruned, leading to a slightly skewed output distribution that deviates from the sole target mode. An alternative solution is to only decide whether to perform the sampling, rather than whether to add it to the predicted tree.

Assuming that all single samplings have the same acceptance rate, the target can be modified as:

$$
\begin{aligned}
T &= \sum p_i = \sum s_i \rho \\
&= P[Path_i] \times \prod_j j < k(1 - draft_j) \times \rho
\end{aligned}
\qquad (7)
$$

where $s_i$ denotes the probability that we make this sampling, and $\rho$ denotes the acceptance rate of a single isolated sampling.

For multi-branch tokens under the same ancestor path, after we sample the first token $t_1$, the second token $t_2$ should never be $t_1$ because it will never pass the verification (The residual probability of target will be zero.). We should only sample the second one from the remaining tokens. Let $d_i$ denote the original output distribution of the draft model, then the probability of sampling the second token $t_2$ can be expressed as $draft_2 = d_{t_2}/(1 - d_{t_1})$.

More generally, for the $k$-th token $t_k$, the probability of sampling it can be calculated as:

$$draft_k = \frac{d_{t_k}}{1 - (\sum_{j<k} d_{t_j})} \qquad (8)$$

Combining the previous formulations, the probability of verifying the $i$-th token given the ancestor $Path_i$, $P[verify_i|Path_i]$, can be expressed as:

$$
\begin{aligned}
P[verify_i|Path_i] &= \prod_{j<i}(1 - draft_j) \\
&= \prod j < i(1 - \frac{d_{t_j}}{1-(\sum_{k<j} d_{t_k})}) \\
&= \prod j < i \frac{1-(\sum_{k<j} d_{t_k})-d_{t_j}}{1-(\sum_{k<j} d_{t_k})} \\
&= 1 - \sum_{j<i} d_{t_j}
\end{aligned}
\qquad (9)
$$

For the probability of the path, $P[path_i]$, where $path_i = x_1, ..., x_{i-1}$, and under the independence assumption, we have:

$$
\begin{aligned}
P[path_i] &= \prod_{j<i} P[accept\, x_j|path_j] \\
&= \prod_{j<i} P[verify_j|Path_j] \times draft_j \\
&= \prod_{j<i}(1 - \sum_{k<j} d_{t_k})\frac{d_{t_j}}{1-\sum_{k<j} d_{t_k}} \\
&= \prod_{j<i} d_{t_j}
\end{aligned}
\qquad (10)
$$

Combining these, the final target expression becomes:

$$
\begin{aligned}
T &= \sum p_i \\
&= \sum_i P[path_i] P[verify_i|Path_i]\rho \\
&= \sum_i \prod_{j \in path_i} d_{t_j} \rho \\
&\quad \times (1 - \sum_{k \text{ is the sibling token before } i} d_{t_k})
\end{aligned}
\qquad (11)
$$

Note that for deeper tokens and sibling tokens after, the acceptance rate $p_i$ will monotonically decrease, which means we can construct the predicted tree greedily.

Our method ensures that at each step, we perform sampling with the maximum expected acceptance rate. To demonstrate this, assume that there exists an alternative method that can generate a better tree of the same size $n$. There must be at least one leaf node that differs between this alternative method and our method. Let's denote the leaf nodes from the alternative method as $N_c$ and the corresponding leaf nodes from our method as $N_{our}$. Furthermore, let's denote the first ancestor node of $N_c$ that is not present in our result as $M_c$, and assume that there are $k$ nodes in the sub-tree of $M_c$.

Denote the expected acceptance rate of this sample as $P[M_c]$. Then, the contribution of the entire sub-tree is at most $k \times P[M_c]$. The fact that our method did not choose this sub-tree implies that the last $k$ samples we made, which are not present in the alternative method, have an expected acceptance rate higher than $P[M_c]$. The contribution of these $k$ samples to the expectation of the total number is larger than $k \times P[M_c]$.

By eliminating these $k$ nodes and applying induction, we can show that $E_{n-k,ours} \geq E_{n-k,c}$, where $E_{n-k,ours}$ and $E_{n-k,c}$ represent the expected number of accepted tokens for our method and the alternative method, respectively. Additionally, we have $\sum^k P[M_{i,ours}] \geq k \times P[M_c] \geq \sum^k P[M_{i',c}]$, where $M_{i,ours}$ and $M_{i',c}$ are the corresponding ancestor nodes in our method and the alternative method, respectively. Combining these results, we can conclude that $E_{n,ours} \geq E_{n,c}$, proving that our method can maximize the expected number of accepted tokens.

### D.1 GREEDY OPTIMAL PROOF

The search space for the responses form a hierarchical $k$-wise tree $S$, with $k$ being the number of tokens in the vocabulary. For a model $M$, it induce a set of weights on the search space. More specifically, for any node $u_n$, assume the unique path starting from the root that lead to $u_n$ is $u_0, u_1, ..., u_n$, define the weight for node $u_n$ to be:

$$w_{u_n} = \Pi_{m=0}^{n-1} P_M(u_{m+1}|u_{0:m}) \tag{12}$$

Consider a subset $S'$ of the space $S$, the weight of the set $w_{S'}$ is defined as the summation of all the nodes' weights in the subset, *i.e.*:

$$w_{S'} = \sum_{v \in S'} w_v \tag{13}$$

Define $\mathcal{T}$ to be the collection of all connected sub-trees that contain the root. We are interested in finding sub-trees with the max weight with number of nodes less than $N$, *i.e.*

$$\mathcal{T}_N^* = \{T | w_T = \max_{T \in \mathcal{T}} w_T\} \tag{14}$$

**Algorithm (Greedy)**: Suppose we start from the set that only contain the root $M_1 = \{root\}$.

Define the candidate set $C(M_i) = N(M_i) \backslash M_i$

Pick the node $v^* = \arg\max_{v \in C(M_i)} w_v$

$M_{i+1} = M_i \cup \{v^*\}$

**Theorem**:

(A) $M_N \in \mathcal{T}$

(B) $M_N \in \mathcal{T}_N^*$

*Proof.* We will prove each part of the theorem separately.

We first prove (A), which is equivalent to verify $M_N$ forms a connected tree that contain the root. The latter fact is trivial since $root \in M_1 \subset M_N$. It's also straightforward to see the connectivity as at every step the new added node belongs to the neighbor. Finally, since a connected subset of a tree $S$ is also a tree, therefore we conclude (A).

For (B), we prove by induction. For $N = 1$, this is trivial. Suppose for $N \leq k$, $M_N \in \mathcal{T}_N^*$, we prove this for $N = k+1$. For any $M'_{k+1} \in \mathcal{T}_{k+1}$, and any $M_k \in \mathcal{T}_k^*$, we show $w_{M_k} + \max_{v \in C(M_k)} w_v \geq w_{M'_{k+1}}$.

To show this, note that $|M'_{k+1}| = k+1 > k = |M_k|$, there exist at least one leaf node $v \in M'_{k+1}$ such that $v \notin M_k$. Consider the unique path that connect the root and $v$ as $u_0, ..., u_p = v$. Since $u_0 \in M_k$ and $u_p \notin M_k$, there must be some $q \in \{1, ..., p\}$ satisfy $u_{q-1} \in M_k$ and $u_q \notin M_k$. By definition, $u_q \in C(M_k)$ since it's the neighbor of $M_k$. And according to the definition of the weight, $w_{u_q} \geq w_{u_p}$. Now consider the fact that $M'_{k+1} \backslash w_{u_p}$ is still a tree since $u_p$ is a leaf, so by induction, we have $w_{M_k} \geq w_{M'_{k+1} \backslash w_{u_p}}$. Therefore, we have

$$
\begin{aligned}
& w_{M_k} + \max_{v \in C(M_k)} w_v \\
\geq\ & w_{M_k} + w_{u_q} \\
\geq\ & w_{M_k} + w_{u_p} \\
\geq\ & w_{M'_{k+1} \backslash w_{u_p}} + w_{u_p} \\
=\ & w_{M'_{k+1}}
\end{aligned}
\tag{15}
$$

Because $M'_{k+1}$ is chosen arbitrarily, we proved that $w_{M_k} + \max_{v \in C(M_k)} w_v = w_{M'_{k+1}}$, completing the proof of (B). $\square$

