# OpenReview forum: "DySpec: Faster Speculative Decoding with Dynamic Token Tree Structure"
_ICLR.cc/2025/Conference — Submitted to ICLR 2025_

### Official Review · Reviewer_Jh6N · 2024-10-27

**Soundness:** 2
**Presentation:** 2
**Contribution:** 2
**Rating:** 3
**Confidence:** 5

**Summary:**

This paper proposes a draft token selection method in speculative decoding aimed at improving the token acceptance rate. The core idea is to use the draft model's prediction score as evidence to infer the token acceptance rate and use this information to select more promising tokens. The authors provide both theoretical and empirical analyses to support their approach.

**Strengths:**

The proposed token selection method demonstrates improvement over existing methods such as Sequoia and Specinfer.

**Weaknesses:**

1. The paper lacks discussion on existing works that share very similar ideas (see Questions 1 and 2).
1. The experimental results do not sufficiently validate the method's effectiveness from various perspectives (see Questions 3 and 4).

**Questions:**

1. There are concurrent works using similar ideas. For instance, EAGLE 2 (https://arxiv.org/abs/2406.16858) attempts to maximize the acceptance rate based on a similar intuition (see Figure 6 in their paper). The authors should mention these works. What are the differences between your method and EAGLE 2?
1. Dynamic tree-based selection methods have appeared in prior literature. For example, Recurrent Drafter (https://arxiv.org/abs/2403.09919) proposes an RNN draft model and uses beam search for adaptive token selection. Did you modify the draft model architecture in your approach?
1. In practice, small draft models may be overconfident about incorrect answers in reasoning tasks. Have you studied your Hypothesis 1 across different tasks and analyzed the failure examples?
1. Does your method support batch inference? Have you experimented with large batch sizes to push your algorithm to its computational limits and test its throughput?

---

> ### Author Response · Authors · 2024-11-21
> **Response to Reviewer Jh6N (1/2)**
>
> We thank the reviewer for his efforts and insightful feedback. Below, we will try to address your questions carefully. We hope the reviewer will consider raising your score in light of our response.
>
> ### **Q1: Regarding related works (EAGLE-2 and ReDrafter).**
>
> Thank you for bringing these methods to our attention. We will add one additional related works section (which was not included previously due to page restriction) to discuss the difference between these methods in detail.
>
> 1. The main difference between EAGLE-2 and DySpec (ours) is that
>     - **Feature-based vs. Model-based:** EAGLE-2 is a **self-speculative** method that makes draft predictions based on the target model's features, rather than a much smaller draft model. Due to the strong drafting capability, self-speculative methods (Medusa, EAGLE, and EAGLE-2) can usually guess with higher accuracy under the same budget. In this work, we mainly focus on **how to build near-optimal draft trees** with a given target and draft, which is orthogonal to **using stronger draft models**. This is also why we do not compare self-speculative methods in our experiments section. Nevertheless, we would like to extend the DySpec tree construction method into self-speculative fashions in future works.
>     - **Tree construction:** EAGLE-2 builds their draft trees with an expand-rerank procedure: first **selects** top-k tokens at each node, and prunes the candidate tree with draft probability. While DySpec **samples** tokens with weights according to Eq (2), which is essentially the probability of **sampling with replacement**. In other words, DySpec **maximizes the sum of expected draft probabilities of all candidate nodes** with a given budget. We also implement a version of **tree construction with threshold** to accelerate sampling from the draft model by reducing sampling times, as shown by Algorithm 2.
>     - **Verification:** One following problem with EAGLE-2 is that it cannot accept tokens by standard verification, i.e. only reject the draft with probability $1 - \frac{target}{draft}$ when $draft > target$, since draft tokens are predicted by **selection rather than sampling**. Instead, EAGLE-2 is verified by sampling a new token from the target at each draft node, and seeing whether it matches the draft token. The problem here is that even in the case that **draft probability is identical to target probability**, the latter verification may yield a low acceptance rate. Imagine a distribution of $[0.25, 0.25, 0.25, 0.25]$ over a vocabulary of $4$. No matter how the draft model guesses, there is only a probability of $0.25$ that it aligns with the target. In contrast, the sampling method adopted by DySpec can yield a perfect acceptance rate of $1$ in this case.
> 2. We also carefully review the tree-based selection method in ReDrafter. We would like to point out that ReDrafter is a **self-speculative** method that uses **independent sampling** in tree construction (similar to SpecInfer). As in DySpec, we use a much smaller draft model rather than target model features to make the predictions. Again, we want to emphasize that our paper concentrates on **building and implementing near-optimal draft trees** with a given target and draft, and we are open to leaving the combination with self-speculation as future works.
>
> In summary, we thank the reviewer for raising these complementary-related works. We will include the discussions in our final version.
>
>
> ### **Q2: The effectiveness of Hypothesis 1 under different domains.**
>
> We totally understand your concerns about the generalization of DySpec. We would like to point out that the effectiveness of DySpec builds on the similar nature of draft distribution and target distribution: it turns out to work fine both theoretically and empirically when the two distributions are "close", according to the discussion in our paper. This is also the case where speculative decoding itself works.
>
> Besides, we also conduct more experiments under practical corpus, i.e. GSM8K (math) and MT-Bench (general chat) to further show the effectiveness of DySpec. The detailed results can be found in our response to Reviewer rcVF, Q2. DySpec consistently outperforms compared baselines, Sequoia and SpecInfer, under both greedy and non-greedy settings across different draft/target sizes. Particularly, under the memory-bound 7b/70b setup, DySpec can hit 50%-60% more draft tokens with greedy decoding and 20%-35% with non-greedy decoding. In terms of speedup, the performance gain is around 45% and 15%-20%, respectively. We believe that **such strong performance across various domains and settings** is able to demonstrate the effectiveness of DySpec's tree construction.

---

> ### Author Response · Authors · 2024-11-21
> **Response to Reviewer Jh6N (2/2)**
>
> ### **Q3: Support for various batch sizes.**
>
> Thank you for this question. We agree with you that tree-based speculative decoding methods give more gain for small batch sizes, where the generation of the target model is more **memory-bound**, thus verifying a larger tree incurs less overhead relatively. This conclusion also aligns with the result that the 7b/70b setting achieves the highest speedup. For larger batch sizes, one can afford less budget for speculation, thus the relative gain of optimal tree structures compared to baselines (e.g. chains) gets smaller. However, it is still possible to use DySpec in real time to push the performance limits at larger batch sizes.

---

> ### Author Response · Authors · 2024-11-25
> **Looking forward to your feedback**
>
> Dear Reviewer,
>
> Thank you for handling our manuscript and providing valuable feedback. We hope that our responses have sufficiently addressed the concerns you raised. We welcome more discussion if you have more questions and suggestions. As the discussion deadline is approaching, we would be very grateful if you could take a moment to review our reply.

---

### Official Review · Reviewer_Dmmc · 2024-10-31

**Soundness:** 3
**Presentation:** 2
**Contribution:** 3
**Rating:** 5
**Confidence:** 4

**Summary:**

Summary:

The authors propose dyspec to generate an optimal tree. The core idea is based on the idea that the target distribution and draft distribution should match.

The authors based on the idea propose an algorithm to come up with optimal tree. They overcome several implementation challenges and achieve reasonable speedups compared to existing baselines.

“DYSPEC achieves a 9.1× throughput improvement and 9.4× reduction in latency.” ([“DySpec”, p. 2]-> Should compare to SpecInfr or Medusa

“Figure 2: Connection between acceptance rate/target distribution and draft distribution on CNN DailyMail.The density of each block is normalized by column.” ([“DySpec”, p. 3] -> Can you please explain this figure. I think it’s the main motivation for your method, however it is not clearly explained, what this figure is showing.

“Specinfer-Baseline” ([“DySpec”, p. 7] -> Specifinfer requires training the models. What have authors done here.

“DYSPEC leverages CUDA Graph to capture 129 different input lengths ranging from 128 to 258” ([“DySpec”, p. 7-> What is the significance of CUDA Graph here, my understanding is CUDA graph is a mechanism to launch multiple kernels at the same time on the GPU to minimize the overhead of kernel dispatch

“We selected Llama2-7B as the draft model” ([“DySpec”, p. 7] -> What happens when you use the 68M model ?

“Set the maximum draft token tree size to 64, DYSPEC achieves up to a 9.1x improvement in throughput and a 9.4x reduction in latency compared to auto-regressive generation” ([“DySpec”, p. 8]-> Unfair comparison, compare to baselines

Additional experiments -

I would really like to experimentally understand the optimality of your tree. Will it be possible to construct an optimal tree based on some post-hoc data and compare it with the tree generated by Dyspec.

**Strengths:**

- The proposed idea is quite useful and can provide significant speedups for tree based speculative decoding methods.

- The overheads reported our negligible.

**Weaknesses:**

- See the summary section please.

**Questions:**

- See the summary section

Please answer some of the question there. I am happy to bump up the score to 6.

---

> ### Author Response · Authors · 2024-11-21
> **Response to Reviewer Dmmc**
>
> Thank you very much for your thoughtful review. We are glad that the reviewer appreciates our **motivation**, **careful implementation**, and **significant speedups**. Below, we will try to address your questions carefully. We hope the reviewer will consider raising your score in light of our response.
> ### **Q1: Inaccuracy in expressions.**
> Thank you for pointing out. We will express more precisely the competitor when comparing the improvement in our abstract and introduction. To clarify, we achieve up to 50% speedup in the 7b/70b setting compared with Sequoia, a strong baseline.
> ### **Q2: More explanation in Figure 2.**
> Section 3 mainly elaborates on our hypothesis of the strong correlation between draft probability and expected acceptance rate. The results in Figure 2 act as empirical evidence for the hypothesis: **tokens with higher draft probability usually have higher target probability as well as higher acceptance rate**.
>
> Specifically, each number in the grid map represents the density w.r.t. column, which is draft probability in both cases. For instance, the top right number $0.95$ in Figure 2a shows that 95% of tokens with draft probability among $(0.8, 1.0]$ also have an acceptance rate among $(0.8, 1.0]$. We believe that such observation provides a strong empirical guarantee for Hypothesis 1 and motivates our algorithm design.
> ### **Q3: Training draft model for SpecInfer.**
> In the original paper, SpecInfer trains JF68m to close the gap between the draft model and the target model (Llama). We follow their setting and use identical draft/target pairs, which are also adopted by Sequoia. Therefore no retraining is needed in this case.
> ### **Q4: The significance of CUDA graphs.**
> In DySpec implementation, we adopt CUDA graphs to accelerate draft model computation by eliminating CPU overheads, which is particularly crucial under the **68m/7b** setting. To minimize the latency of the draft model, we capture a graph for each input length, which requires 128 captures to support generating 128 tokens in our experimental setup.
>
> When the draft model's latency is already sufficiently smaller than the target model's (i.e. the **7b/70b** setting), the overhead of the draft model becomes less significant. Consequently, even without CUDA graph acceleration, a considerable end-to-end speedup ratio can still be achieved.
>
> Thank you for raising this concern. We will clarify more on the usage of the CUDA graph in our final revision.
> ### **Q5: 7b vs. 68m on 70b target model.**
> Following our discussion in Q4, since the 70b target model has a sufficiently larger latency compared with the 7b draft model (as shown by Figure 4c), further replacing it with a 68m size becomes less significant. In fact, the size of the draft model is a "latency-accuracy" trade-off. In this case, the 7b model yields a much higher acceptance rate at the cost of a slightly higher draft/target latency ratio, overall resulting in better performance.
> ### **Q6: About optimality.**
> Thank you for your question. We would like to explain this problem in detail.
>
> First, we want to clarify that DySpec and Sequoia employ a **sampling with replacement** in tree construction, while EAGLE-2 uses **selection and pruning**. During verification, one following problem of EAGLE-2 is that it cannot accept tokens by standard verification, i.e. only reject the draft with probability $1 - \frac{target}{draft}$ when $draft > target$, since draft tokens are predicted by **selection rather than sampling**. Instead, EAGLE-2 is verified by sampling a new token from the target at each draft node, and seeing whether it matches the draft token. The problem here is that even in the case that **draft probability is identical to target probability**, the latter approach may yield a low acceptance rate. Imagine a distribution of $[0.25, 0.25, 0.25, 0.25]$ over a vocabulary of $4$. No matter how the draft model guesses, there is only a probability of $0.25$ that it aligns with the target. In contrast, the sampling method adopted by DySpec yields a perfect acceptance rate of $1$ in this case.
>
> Back to the cases of DySpec and Sequoia. Since the trees are built by sampling, it is impossible to find a **fixed** optimal tree in advance. We can only compute and compare the expected accepted tokens once the tree is sampled and determined. Nevertheless, we present a case study on initial guesses of DySpec and Sequoia in [Figure](https://anonymous.4open.science/r/rbt-56E3/comparison.jpg). It shows a vocabulary space with depth and width both set to 3. Nodes chosen by DySpec are colored in blue and Sequoia are colored in orange. As shown, at the divergent step, DySpec will not sample from the node "com" with an acceptance rate of $0$ (since $target=0$) due to its low weight (draft probability). Instead, it samples the node "The" with an acceptance rate of 1 (since $draft < target$). In other words, DySpec can better escape from the local minimum and fully utilize the speculation budget.

---

> ### Author Response · Authors · 2024-11-25
> **Looking forward to your feedback**
>
> Dear Reviewer,
>
> Thank you for handling our manuscript and providing valuable feedback. We hope that our responses have sufficiently addressed the concerns you raised. We welcome more discussion if you have more questions and suggestions. As the discussion deadline is approaching, we would be very grateful if you could take a moment to review our reply.

---

### Official Review · Reviewer_SL3U · 2024-11-03

**Soundness:** 3
**Presentation:** 2
**Contribution:** 2
**Rating:** 6
**Confidence:** 2

**Summary:**

This paper introduces DySpec, a novel approach to speculative decoding that employs a dynamic token tree structure to improve inference speed for large language models. The authors present both theoretical and empirical evidence showing that higher draft probabilities correlate with higher acceptance rates. Based on this insight, they develop a greedy strategy for dynamically expanding the token tree at runtime. The method achieves impressive results, demonstrating up to 9.1× throughput improvement and 9.4× reduction in latency on Llama2-70B under low temperature settings, outperforming existing methods like Specinfer and Sequoia.

**Strengths:**

1. Strong Theoretical Foundation
- Provides rigorous theoretical analysis linking draft distribution to acceptance rate
- Includes formal proofs of optimality under stated assumptions
- Clearly bridges theoretical insights with practical implementation

2. Novel Technical Contributions
- Introduces an innovative dynamic token tree construction approach
- Develops efficient algorithms for both fixed-size and threshold-based tree construction
- Proposes block-sparsity friendly token ordering for optimization

3. Comprehensive Empirical Evaluation
- Tests across multiple model scales (7B to 70B parameters)
- Evaluates on diverse datasets (C4, OpenWebText, CNN DailyMail)
- Compares against strong baselines (Specinfer, Sequoia)
- Examines performance under different temperature settings

4. Implementation Efficiency
- Addresses practical concerns about overhead
- Provides C++ implementation to minimize token tree construction costs
- Includes detailed analysis of computation complexity

**Weaknesses:**

1. Limited Discussion of Limitations
- Could elaborate more on scenarios where the method might not perform optimally
- More discussion of the trade-offs between fixed-size and threshold-based approaches would be valuable

2. Implementation Details
- Some implementation specifics about the C++ optimizations could be expanded
- Could provide more guidance on threshold selection for different scenarios

3. Experimental Validation
- Could include more ablation studies to isolate the impact of different components
- Additional experiments on more diverse model architectures would strengthen generalizability claims

**Questions:**

None

---

> ### Author Response · Authors · 2024-11-21
> **Response to Reviewer SL3U**
>
> Thank you very much for your thoughtful review. We are glad the reviewer found our work **novel**, **practical**, and **theoretically and empirically sound**.
> Below, we will try to address your questions carefully. We hope the reviewer will consider raising your score in light of our response.
>
> ### **Q1: Discussion of limitations.**
>
> Thank you for this question. In short, DySpec has the same bottleneck as most speculative methods:
> - The performance is bounded by how similar draft and target distributions are. In particular, an accurate draft can build a near-optimal draft tree at the tree construction stage and accept more tokens at the verification stage.
> - The method has a larger gain in memory-bound scenarios. As shown, the 7b/70b setting has the highest speedup and drops as the generation turns to computation-bound. The reason is that verifying a larger tree incurs less overhead relatively in memory-bound scenarios, and the target model benefits more from near-optimal draft trees provided by DySpec. Nevertheless, we would like to point out that other prevalent speculative decoding methods, e.g. Sequoia and SpecInfer, also share the same trend.
>
> ### **Q2: Trade-offs between fix-size and threshold-based implementations.**
>
> Thank you for this question. Given the optimization objective of accepted tokens, it is clear that the fix-size algorithm, which expands the tree greedily, yields the best returns. However, in implementation, the fix-size algorithm may incur many calls of draft model generation, which is inefficient (since we do not know which leaf node to expand from each time). Therefore, we propose the threshold-based algorithm to pack the calls of the draft model into batches to reduce the number of passes. We also incorporate an early-exit mechanism to further improve efficiency: when the weight of every leaf node (i.e. the expected accepted probabilities) is below the threshold, we stop generating new tokens and verify the current tree with the target model. In short, the threshold-based algorithm is a practical approximation of the fixed-sized algorithm with higher efficiency and less execution time in the real world.
>
>
> ### **Q3: More details in C++ optimizations.**
>
> The overhead introduced by DySpec can be primarily attributed to draft tree construction. Even in the faster threshold-based algorithm, the tree expansion remains time-consuming due to its complex logic. To mitigate this, we maintain the tree structure with C++ implementation, which includes adding new nodes, generating input positions, and creating tree attention masks (here we also use PyTorch's C++ API to optimize the retrieval of parents' indices). As a result, we finally reduced the tree construction time by more than 50% compared to vanilla Python implementation.
>
> ### **Q4: Experimental validations.**
>
> We understand the reviewer's concerns about the effectiveness of our different components in implementation, and whether it can generalize to other model architectures.
> - For the first question, we would like to point out that our dynamic tree algorithm is rather monolithic, and we walk through various optimization techniques to eliminate and overlap the additional cost induced by building the dynamic tree, which serves the same purpose. Therefore, we do not conduct much performance breakdown for these optimization tricks.
> - For the latter question, we would like to kindly remind that we have included various range of model sizes (68m/7b, 68m/13b, 7b/70b) in our experiments, and this paper mainly focuses on accelerating the generation of transformer models. Extending to other model architectures (e.g. Mamba) is interesting but beyond the scope. In cases where the reviewer has concerns about generalization to other domains, we also add additional experiments on GSM8K and MT-Bench, two common evaluation benchmarks, which can be found in our response to Reviewer rcVF, Q2. We hope that the reviewer finds these results helpful.

---

> ### Author Response · Authors · 2024-11-25
> **Looking forward to your feedback**
>
> Dear Reviewer,
>
> Thank you for handling our manuscript and providing valuable feedback. We hope that our responses have sufficiently addressed the concerns you raised. We welcome more discussion if you have more questions and suggestions. As the discussion deadline is approaching, we would be very grateful if you could take a moment to review our reply.

---

### Official Review · Reviewer_rcVF · 2024-11-04

**Soundness:** 4
**Presentation:** 2
**Contribution:** 3
**Rating:** 5
**Confidence:** 4

**Summary:**

This work proposes a method to dynamically expand the token tree based on the draft distribution. A key difference between this work and existing approaches is that it introduces a smooth dynamic draft token tree construction method, which expands the tree along both width and depth without hyperparameter setup, within the given token budget. However, related works are missing, and the experimental setup should be properly improved.

**Strengths:**

1.	The correlation of Hypothesis 1 with the proposed method is well elaborated and explained.
2.	The structure of the paper is clear and easy to follow.

**Weaknesses:**

1.	Lack of related work. Context-aware dynamic draft token tree is not a new idea. I would like to draw your attention to a very related work: “EAGLE-2: Faster Inference of Language Models with Dynamic Draft Trees” (EMNLP'24). This paper also proposes adopting dynamic draft token trees and claims a strong positive correlation between the draft model confidence score and the acceptance rate of the token. Another relevant paper is “Dynamic Depth Decoding: Faster Speculative Decoding for LLMs,” which further improves performance by dynamic depth. The methodology of EAGLE-2 is quite similar to this paper but differs in the tree construction method. Including this work in your paper and providing necessary discussion is essential.
2.	Experimental setup and results are weak. First, the dataset is too limited. I recommend adding more datasets like MT-Bench (Zheng et al., 2023), HumanEval (Chen et al., 2021), and GSM8K (Cobbe et al., 2021). Second, the 9.1× speedup is compared with the autoregressive method, and when compared with static tree methods like Sequoia, the results are much less overwhelming. Considering that the experimental setup lacks comparison with state-of-the-art dynamic draft token tree methods, the experiment does not fully convince me of the merits of this methodology.
3.	Writing needs improvement. Section 4.2 is hard to follow. The presentation of Figure 3 should be improved. Additionally, Figure 4 is not clear; please increase the font size.

**Questions:**

More convincing experimental results that compare with more relevant related work.

---

> ### Author Response · Authors · 2024-11-21
> **Response to Reviewer rcVF (1/2)**
>
> Thank you very much for your thoughtful review. We are glad that the reviewer found our work **well motivated** and **easy to follow**.
> Below, we will try to address your questions carefully. We hope the reviewer will consider raising your score in light of our response.
>
> ### **Q1: Regarding related works (EAGLE-2 and Dynamic Depth Decoding).**
>
> Thank you for bringing these methods to our attention! We will add one additional related works section (which was not included previously due to page restriction) to discuss the difference between these methods in detail.
>
> The main difference between EAGLE-2 and DySpec (ours) is that
> - **Feature-based vs. Model-based:** EAGLE-2 is a **self-speculative** method that makes draft predictions based on the target model's features, rather than a much smaller draft model. Due to the strong drafting capability, self-speculative methods (Medusa, EAGLE, and EAGLE-2) can usually guess with higher accuracy under the same budget.
> In this work, we mainly focus on **how to build near-optimal draft trees**
> with a given target and draft, which is orthogonal to **using stronger draft models**. This is also why we do not compare self-speculative methods in our experiments section. Nevertheless, we would like to extend the DySpec tree construction method into self-speculative fashions in future works.
> - **Tree construction:** EAGLE-2 builds their draft trees with an expand-rerank procedure: first **selects** top-k tokens at each node, and prunes the candidate tree with draft probability. While DySpec **samples** tokens with weights according to Eq (2), which is essentially the probability of **sampling with replacement**. In other words, DySpec **maximizes the sum of expected draft probabilities of all candidate nodes** with a given budget. We also implement a version of **tree construction with threshold and early exit** to accelerate sampling from the draft model by reducing sampling times, as shown by Algorithm 2.
> - **Verification:** One following problem with EAGLE-2 is that it cannot accept tokens with standard verification, i.e. only reject the draft with probability $1 - \frac{target}{draft}$ when $draft > target$, since draft tokens are predicted by **selection rather than sampling**. Instead, EAGLE-2 is verified by sampling a new token from the target at each draft node, and seeing whether it matches the draft token. The problem here is that even in the case that **draft probability is identical to target probability**, the latter verification may yield a low acceptance rate. Imagine a distribution of $[0.25, 0.25, 0.25, 0.25]$ over a vocabulary of $4$. No matter how the draft model guesses, there is only a probability of $0.25$ that it aligns with the target. In contrast, the sampling method adopted by DySpec can yield a perfect acceptance rate of $1$ in this case.

---

> ### Author Response · Authors · 2024-11-21
> **Response to Reviewer rcVF (2/2)**
>
> ### **Q2: Extending experiment settings.**
> We totally understand the reviewer's concerns about experiment setups. We add experiment results on more corpus to thoroughly show the performance comparison between DySpec and baselines.
>
> (*Note: HumanEval is not included here because the draft model, JF68m, is only trained on Wikipedia and C4-en, with no coding corpus.*)
>
> - **68m/7b**: **Speedup** to auto-regressive / **Accepted tokens** with a budget of 64
>
>     | Dataset   | Temperature   | DySpec                | Sequoia       | SpecInfer     |
>     | :-:       | :-:           | :-:                   | :-:           | :-:           |
>     | GSM8K     | 0.0           | **3.93** / **6.86**   | 2.79 / 4.92   | 2.01 / 3.47   |
>     | GSM8K     | 0.6           | **2.44** / **4.31**   | 2.20 / 3.55   | 1.65 / 3.03   |
>     | MT-Bench  | 0.0           | **2.59** / **4.02**   | 2.35 / 3.55   | 1.68 / 2.70   |
>     | MT-Bench  | 0.6           | **2.15** / **3.62**   | 2.11 / 3.18   | 1.50 / 2.71   |
> - **68m/13b**: **Speedup** / **Accepted tokens**
>
>     | Dataset   | Temperature   | DySpec                | Sequoia       | SpecInfer     |
>     | :-:       | :-:           | :-:                   | :-:           | :-:           |
>     | GSM8K     | 0.0           | **3.17** / **5.29**   | 2.21 / 3.92   | 1.74 / 2.98   |
>     | GSM8K     | 0.6           | **2.49** / **4.17**   | 2.10 / 3.39   | 1.51 / 2.72   |
>     | MT-Bench  | 0.0           | **2.19** / **3.72**   | 2.15 / 3.46   | 1.50 / 2.62   |
>     | MT-Bench  | 0.6           | **2.25** / **3.62**   | 1.93 / 3.11   | 1.40 / 2.60   |
> - **7b/70b**: **Speedup** / **Accepted tokens**
>
>     | Dataset   | Temperature   | DySpec                | Sequoia       | SpecInfer     |
>     | :-:       | :-:           | :-:                   | :-:           | :-:           |
>     | GSM8K     | 0.0           | **10.56** / **12.39** | 7.31 / 7.62   | 5.22 / 5.34   |
>     | GSM8K     | 0.6           | **7.57** / **8.14**   | 6.62 / 6.89   | 5.75 / 5.89   |
>     | MT-Bench  | 0.0           | **9.95** / **11.25**  | 6.96 / 7.46   | 4.75 / 4.85   |
>     | MT-Bench  | 0.6           | **8.47** / **10.11**  | 6.96 / 7.46   | 5.52 / 5.67   |
>
> In summary, DySpec consistently outperforms compared baselines, Sequoia and SpecInfer, under both greedy and non-greedy settings across different draft/target sizes. Compared with Sequoia, the relative speedup can be up to 45% and the average acceptance rate can be up to 63%.
>
> As we mentioned in Q1, we do not compare with EAGLE-2 due to the unfair nature of self-speculative methods. The combination of DySpec's tree construction and self-speculative methods is beyond the scope of this paper, and we are looking forward to addressing it in future works.
>
>
> ### **Q3: Writing issues.**
>
> Thank you for pointing out our writing issues. We will polish our algorithm section ($\S$ 4.2) and refine the presentation of Figure 3 and Figure 4. We believe that we will present a better final version through revision.

---

> ### Author Response · Authors · 2024-11-25
> **Looking forward to your feedback**
>
> Dear Reviewer,
>
> Thank you for handling our manuscript and providing valuable feedback. We hope that our responses have sufficiently addressed the concerns you raised. We welcome more discussion if you have more questions and suggestions. As the discussion deadline is approaching, we would be very grateful if you could take a moment to review our reply.

---

### Author Response · Authors · 2024-11-21
**General Response to Reviewer and AC**

We thank all reviewers and AC for their efforts in reviewing and valuable feedback. We notice that many reviewers raise concerns about DySpec experiment coverage (rcVF, SL3U, Jh6N) and differences with some particular tree-based speculative decoding methods, e.g. EAGLE-2 (rcVF, Jh6N). Below we will address these common concerns and we hope that reviewers find our response well-clarified and helpful.

### **1. Comparison with EAGLE-2.**

As a concurrent work, the main difference between EAGLE-2 and DySpec (ours) is that
- **Feature-based vs. Model-based:** EAGLE-2 is a **self-speculative** method that makes draft predictions based on the target model's features, rather than a much smaller draft model. Due to the strong drafting capability, self-speculative methods (Medusa, EAGLE, and EAGLE-2) can usually guess with higher accuracy under the same budget.
In this work, we mainly focus on **how to build near-optimal draft trees**
with a given target and draft, which is orthogonal to **using stronger draft models**. This is also why we do not compare self-speculative methods in our experiments section. Nevertheless, we would like to extend the DySpec tree construction method into self-speculative fashions in future works.
- **Tree construction:** EAGLE-2 builds their draft trees with an expand-rerank procedure: first **selects** top-k tokens at each node, and prunes the candidate tree with draft probability. While DySpec **samples** tokens with weights according to Eq (2), which is essentially the probability of **sampling with replacement**. In other words, DySpec **maximizes the sum of expected draft probabilities of all candidate nodes** with a given budget. We also implement a version of **tree construction with threshold and early exit** to accelerate sampling from the draft model by reducing sampling times, as shown by Algorithm 2.
- **Verification:** One following problem with EAGLE-2 is that it cannot accept tokens with standard verification, i.e. only reject the draft with probability $1 - \frac{target}{draft}$ when $draft > target$, since draft tokens are predicted by **selection rather than sampling**. Instead, EAGLE-2 is verified by sampling a new token from the target at each draft node, and seeing whether it matches the draft token. The problem here is that even in the case that **draft probability is identical to target probability**, the latter verification may yield a low acceptance rate. Imagine a distribution of $[0.25, 0.25, 0.25, 0.25]$ over a vocabulary of $4$. No matter how the draft model guesses, there is only a probability of $0.25$ that it aligns with the target. In contrast, the sampling method adopted by DySpec can yield a perfect acceptance rate of $1$ in this case.

### **2. Experiments with more settings.**

We totally understand the reviewer's concerns about experiment setups. We add experiment results on more corpus to thoroughly show the performance comparison between DySpec and baselines.

- **68m/7b**: **Speedup** to auto-regressive / **Accepted tokens** with a budget of 64

    | Dataset   | Temperature   | DySpec   | Sequoia | SpecInfer     |
    | :-:       | :-:           | :-:             | :-:           | :-:           |
    | GSM8K     | 0.0           | **3.93** / **6.86**   | 2.79 / 4.92   | 2.01 / 3.47   |
    | GSM8K     | 0.6           | **2.44** / **4.31**   | 2.20 / 3.55   | 1.65 / 3.03   |
    | MT-Bench  | 0.0           | **2.59** / **4.02**   | 2.35 / 3.55   | 1.68 / 2.70   |
    | MT-Bench  | 0.6           | **2.15** / **3.62**   | 2.11 / 3.18   | 1.50 / 2.71   |
- **68m/13b**: **Speedup** / **Accepted tokens**

    | Dataset   | Temperature   | DySpec                | Sequoia       | SpecInfer     |
    | :-:       | :-:           | :-:        | :-:      | :-:    |
    | GSM8K     | 0.0           | **3.17** / **5.29**   | 2.21 / 3.92   | 1.74 / 2.98   |
    | GSM8K     | 0.6           | **2.49** / **4.17**   | 2.10 / 3.39   | 1.51 / 2.72   |
    | MT-Bench  | 0.0           | **2.19** / **3.72**   | 2.15 / 3.46   | 1.50 / 2.62   |
    | MT-Bench  | 0.6           | **2.25** / **3.62**   | 1.93 / 3.11   | 1.40 / 2.60   |
- **7b/70b**: **Speedup** / **Accepted tokens**

    | Dataset   | Temperature   | DySpec    | Sequoia       | SpecInfer     |
    | :-:       | :-:           | :-:                   | :-:       | :-:     |
    | GSM8K     | 0.0           | **10.56** / **12.39** | 7.31 / 7.62   | 5.22 / 5.34   |
    | GSM8K     | 0.6           | **7.57** / **8.14**   | 6.62 / 6.89   | 5.75 / 5.89   |
    | MT-Bench  | 0.0           | **9.95** / **11.25**  | 6.96 / 7.46   | 4.75 / 4.85   |
    | MT-Bench  | 0.6           | **8.47** / **10.11**  | 6.96 / 7.46   | 5.52 / 5.67   |

In summary, DySpec consistently outperforms compared baselines, Sequoia and SpecInfer, under both greedy and non-greedy settings across different draft/target sizes. Compared with Sequoia, the relative speedup can be up to 45% and the average acceptance rate can be up to 63%.

---

### Author Response · Authors · 2024-11-23
**Looking forward to the discussion**

Dear Reviewers,

We sincerely appreciate your time and effort in reviewing our work. We fully understand your schedule may be quite busy right now. As the deadline for the Author-Reviewer discussion period is approaching, we would greatly value the opportunity to engage in further discussion with you. We look forward to your feedback on whether our responses effectively address your concerns and if there are any additional questions or points you would like to discuss.

Thank you again for your thoughtful consideration.

Best regards,
Authors

---

### Meta-Review · Area_Chair_JhVt · 2024-12-23

**Metareview:**

This paper provides a dynamic tree structure construction for multi-draft speculative decoding, and shows improvements over existing static tree construction techniques. However, during the rebuttal, the reviewers mentioned the existence of related work that also creates trees dynamically, specifically Eagle 2, which has significant overlap with the current proposal. The authors have responded with an explanation on the differences. However, the AC finds this to be insufficient and a more in-depth understanding of similarities and differences is warranted. In particular, the AC recommends that the authors replace the tree construction step in Eagle 2 with that proposed herein. This ablation is key to substantiating the claim of almost optimality of the tree construction in this work and can increase its impact further. We hope that the authors find the comments of the reviewers useful for a future iteration of their paper.

**Additional Comments On Reviewer Discussion:**

The reviewers' main criticism is the overlap with prior work (Eagle 2.0) in terms of conceptual proposal. While I agree with the authors that there are key differences between the two proposals, it is important to also verify the claim of almost optimality of the proposed tree construction on the setup of Eagle 2.0.

---

### Decision · Program_Chairs · 2025-01-22

Reject